# Get More at Once: Alternating Sparse Training with Gradient Correction

**Li Yang**\*, **Jian Meng**\*, **Jae-Sun Seo, Deliang Fan**
School of Electrical, Computer and Energy Engineering
Arizona State University, Tempe, AZ 85287
{lyang166,jmeng15,jseo28,dfan}@asu.edu

## Abstract

Recently, a new trend of exploring training sparsity has emerged, which removes parameters during training, leading to both training and inference efficiency improvement. This line of works primarily aims to obtain a single sparse model under a pre-defined large sparsity ratio. It leads to a static/fixed sparse inference model that is not capable of adjusting or re-configuring its computation complexity (i.e., inference structure, latency) after training for real-world varying and dynamic hardware resource availability. To enable such run-time or post-training network morphing, the concept of 'dynamic inference' or 'training-once-for-all' has been proposed to train a single network consisting of multiple sub-nets once, but each sub-net could perform the same inference function with different computing complexity. However, the traditional dynamic inference training method requires a joint training scheme with multi-objective optimization, which suffers from very large training overhead. In this work, for the first time, we propose a novel alternating sparse training (AST) scheme to train multiple sparse sub-nets for dynamic inference without extra training cost compared to the case of training a single sparse model from scratch. Furthermore, to mitigate the interference of weight update among sub-nets without losing the generalization of optimization, we propose *gradient correction* within the inner-group iterations to reduce their weight update interference. We validate the proposed AST on multiple datasets against state-of-the-art sparse training methods, which shows that AST achieves similar or better accuracy, but only needs to train once to get multiple sparse sub-nets with different sparsity ratios. More importantly, comparing with the traditional joint training based dynamic inference training methodology, the large training overhead is completely eliminated without affecting the accuracy of each sub-net. Code is available at https://github.com/mengjian0502/AST.

## 1 Introduction

For Deep Neural Networks (DNNs), sparsification (i.e., pruning) has been widely explored in the last decade aiming to reduce the computational and memory cost by removing unimportant parameters. Early works of DNN pruning focused on exploring the post-training sparsity [10, 7, 11] to improve the inference efficiency, which prunes a well-trained dense model followed by additional fine-tuning to recover accuracy. Recently, exploring in-training sparsity has emerged as a promising technique to improve the training efficiency by pruning parameters during training [3, 4, 9, 19, 14]. The success of these in-training sparsification methods is mainly rewarded from the *prune-and-regrow* technique, which periodically and aggressively eliminates the unimportant non-zero weights from the

---

\*These authors contributed equally to this work

36th Conference on Neural Information Processing Systems (NeurIPS 2022).

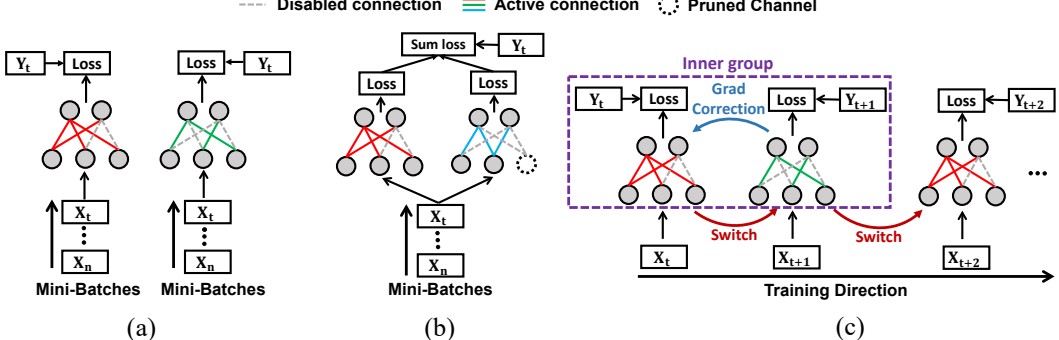

Figure 1: (a) Individually train multiple sub-nets from scratch. (b) Jointly train multiple sub-nets for dynamic inference. (c) The proposed alternating sparse training (AST) scheme.

model, and then regrows certain portion of the pruned candidates back. Until now, such in-training sparsity works primarily focused on training a single sparse model under a pre-defined sparsity ratio, generating a static and fixed sparse network. In other words, the sparsity ratio has to be pre-calculated before training depending on the target hardware resources and application requirement. However, obtaining multiple sparse models (towards storage/energy vs. accuracy trade-off) from the same model architecture requires extra fine-tuning or training overhead.

To enable run-time or post-training network morphing, the concept of 'dynamic inference' or 'training-once-for-all' has emerged recently [24, 2]. Slimmable Neural Network (S-Net) [24] and its optimized counterpart (US-Net) [22] allow the model to switch between different architectures among the pre-defined sub-net candidates with different channel widths. Such run-time dynamics allows users to tune the tradeoffs between model complexity and inference accuracy. To mitigate the weight update interference, S-Net/US-Net forces the smaller sub-net to be completely nested by the larger sub-net. Based on that, OFA [2] and BigNAS [23] are extended to support a much larger number of sub-nets across more dimensions (e.g., depth, width, kernel size, and resolution). However, all prior methods rely on a joint training scheme with multi-objective optimization, which requires minimizing the loss of all sub-nets collectively. Specifically, all sub-nets have to perform forward and backward passes in each mini-batch iteration, resulting in much larger training overhead compared to the individual network training.

To address these concerns, we propose a novel *alternating sparse training* (AST) method for dynamic inference which aims to train multiple sparse sub-nets at one training procedure without extra training cost, compared to training a single sparse model independently [3, 4, 9, 19, 14]. To generate and sparsely train each sub-net, we adapt the *prune-and-regrow* scheme from the in-training sparsity method [14], as it achieves high training sparsity among other schemes. AST alternately trains sub-nets with different sparsity values for different mini-batch iterations, as illustrated in Figure 1. Inspired by the implicit regularization of SGD that was revealed by [18, 16], we further demonstrate such alternating training scheme for dynamic inference helps to maximize the inner product of gradients between the consecutive mini-batch iterations (i.e., sub-nets in our case). However, even with the help of the implicit regularization, we find that negative inner products still remain, which represents the conflicting gradient direction between each two sub-nets. Such phenomenon causes the interference of weight update, while it is helpful to escape from local minima and improve generalization of optimization for individual network training by mini-batch SGD [6, 1]. To mitigate the interference of weight updates between sub-nets but without losing the generalization, we further propose *gradient correction* to remove the conflicting gradient direction between sub-nets only within inner-group iterations, which is defined as $N$ consecutive mini-batch iterations, where $N$ is the number of sub-nets (e.g., $N = 2$ as shown in Figure 1).

Overall, compared to the sparse training and dynamic inference works, AST achieves both training efficiency with sparsity (against dynamic inference) and inference dynamics with multiple sparse sub-nets (against individual sparse training). We conduct comprehensive experiments on CIFAR-10, CIFAR-100 and ImageNet datasets with various DNN models and sparsity granularities. Compared to the SoTA sparse training on CIFAR, the proposed AST algorithm achieves similar or better accuracy while training multiple sparse sub-nets all at once (reduced training cost). Furthermore, compared to the dynamic inference method, AST consistently achieves better accuracy on all the experiments.

## 2 Related Works and Background

### 2.1 Sparse Training

As summarized in [8], sparse training works mainly can be categorized into three groups according to the time point the sparsity is applied: 1) Post-training sparsification [3, 4, 9, 19, 14], which removes the weights by fine-tuning based on a pre-trained model; 2) Before-training [11, 20] sparsification, which obtains a sparse model before the main training procedure; and 3) In-training sparsification, which removes weights during the training process from scratch [14, 26, 4, 3].

**In-training sparsification**    Different from post-training and before-training sparsification, exploiting sparsity during training only requires a single training process from scratch. The pruning topology is gradually perfectized along with the weights optimization. Compared to the before-training sparsification, pruning the model during training makes the gradient visible to the algorithms, so the reflection of the gradient can be used to correct the final pruning decision, leading to better accuracy compared to post-training. Motivated by this, the *prune-and-regrow* technique [3] periodically removes the unimportant non-zero weights from the sparse model and regrows certain pruned weights back during each mini-batch iteration of training process. As a representative work, RigL [4] first prunes a certain ratio $r$ based on weight magnitude as:

$$w^{'} = TopK(|w|, s - r), \tag{1}$$

where $TopK(v, k)$ returns the weight/activation tensor retaining the top $k$-proportion of elements from $v$, and $s$ is the targeted sparsity ratio. After that, the $r$ proportion of new connections are re-generated based on the gradient magnitude in the same mini-batch iteration:

$$w = w^{'} + TopK(g_{i!=w', s+r}) \tag{2}$$

By doing so, such prune-and-regrow scheme optimizes the sparse connection with a fixed sparsity ratio $r$ during the entire training process. Based on this, recent works develop different weight importance criterion to perform prune-and-regrow. SNFS [3] uses the momentum magnitude of weight to do pruning. MEST [26] uses the sum of weight and gradient magnitude to indicate the importance of weights for pruning, and further randomly select weights to grow back. In addition, GraNet [14] employs the same rule as Rigl but adopts a smaller sparsity as initialization (e.g., 0.5) and gradually prunes the model to the target sparsity (e.g., 0.8, 0.9) by following a cosine decay. In addition, more similar to our work, [19, 16] propose to alternately train a dense model and one of its sparse variant (i.e., sub-net) returning both an accurate dense model and a sub-net. However, the rationality behind it is not clearly described. Overall, the objective of our work is obtaining multiple sparse sub-networks with the same amount of training effort as the individual network sparsification. More recently, based on the in-training sparsification mechanism, [13] proposes dynamic sparse training ensemble method to independently generate multiple sparse sub-nets for ensemble. Such method is orthogonal to our work which could be further combined to improve accuracy.

### 2.2 Joint Training for Dynamic Inference

Dynamic inference works [24, 22, 23, 2, 21] primarily aim to train a single network which consists of multiple sub-nets to perform inference independently. Since the weights of the sub-nets are partially shared between each other, they can be switched at run-time permitting dynamic inference accuracy-complexity trade-off. Such concept is firstly proposed by Slimmable Neural Network (S-Net) [24] and its optimized counterpart (US-Net) [22], which train a single neural network executable at different channel-widths. Inspired by S-Net, OFA [2] and BigNAS [23] are proposed to construct a dynamic DNN that includes a much larger number of sub-nets across many more dimensions (e.g., depth, width, kernel size, and resolution) . Note that, as revealed by [2], to enable sub-nets switching with un-compromised individually inference accuracy, the structures of sub-nets are defined by a *subset rule*: the smaller sub-net must be the completely subset of the larger sub-net. Specifically, this line of works train a network in a joint training fashion, which can be expressed as:

$$\min \ \mathbb{E}_W \left( \sum_{i=1}^{N} \mathcal{L}(f(X, \{W_i\}); Y), \right) \tag{3}$$

where $X$ is the mini-batch of inputs with corresponding targets $Y$, and $N$ is the number of sub-nets. $\mathcal{L}(\cdot; \cdot)$ calculates the cross-entropy loss of DNN output and target. $f(X, \{W_i\})$ computes the output of sub-net parameterized by $\{W_i\}$. However, such method consumes much larger training time compared to training a single individual model, since all sub-nets have to perform forward and backward once in each mini-batch iteration as shown in Eq. 3.

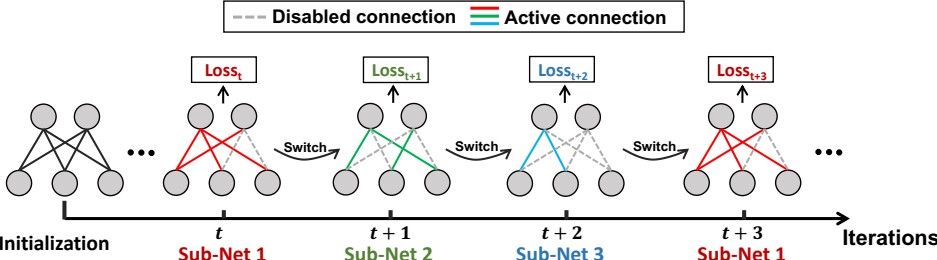

Figure 2: Alternating sparse training (AST): The subset network (sub-net) are iteratively activated throughout the training. The model only learns the active connections of each sub-net, leading to the one-time-trained multiple sub-nets.

## 3 Sparse Train Once Get More

### 3.1 Rationale of Alternating Sparse Training (AST)

Different from joint training for dynamic inference such as S-Net [24] and US-Net [22], which updates all sub-nets in each mini-batch iteration, we propose the alternating sparse training (AST) scheme which trains multiple sub-nets in an alternating fashion over time, where a single sub-net is trained in one iteration. Note that, the sub-net is defined a part weight of the network, which can run inference independently. All the sub-nets are partial shared with each other within a single network. As shown in Figure 2, assuming we have three sub-nets with different sparsity ratio (e.g., sub-net 1 < sub-net 2 < sub-net 3), such training scheme will start to train the sub-net 1 in the first iteration followed by the training of the sub-net 2, and sub-net 3 in the second and third iteration, respectively. It only updates the active connections while ignoring the disabled weights (i.e., pruned for current sparse sub-net), then repeatedly switches sub-nets every three consecutive iterations until the training completes.

The rationale behind AST is inspired by the findings in Reptile [18], which is a meta-learning algorithm originally aiming to learn an initialization of a model from multiple tasks for fast adaption on new tasks. It shows that mini-batch SGD imposes an implicit regularization to maximize the dot-product of consecutive mini-batches. In the following, we will show how this proof adapts to our case. Assuming that we alternately train two sub-nets, sub-net 1 and sub-net 2 with two consecutive mini-batches $B_1$ and $B_2$ respectively, according to the Taylor Series, the gradient of sub-net 2 $g_2$ calculated by SGD can be expressed by:

$$
\begin{aligned}
g_2 = \mathcal{L}'(w_2) &= \mathcal{L}'(w_1) + \mathcal{L}''(w_1)(w_2 - w_1) + O(||w_2 - w_1||^2) \\
&= \mathcal{L}'(w_1) + \overline{H}_1(w_2 - w_1) + O(\alpha^2) \\
&= \mathcal{L}'(w_1) - \alpha\overline{H}_1 g_2 + O(\alpha^2) \quad \text{(using } w_2 - w_1 = \alpha g_2)
\end{aligned}
$$

Where $\overline{H}_1$ is Hessian of the sub-net 1 and $\alpha$ is the current learning rate. Similar to Reptile, the term $\alpha\overline{H}_1 g_2$ serves to maximize the dot-product of the consecutive sub-nets, where the expectation can be expressed as:

$$
\mathbb{E}_{1,2}[\alpha\overline{H}_1 g_2] = \frac{1}{2}\mathbb{E}_{1,2}[\frac{\partial}{\partial w_1}(g_1 \cdot g_2)] \tag{4}
$$

Thus, $-\alpha\overline{H}_1 g_2$ is the direction that serves to maximize the inner product of two consecutive mini-batches, which means to improve the gradient alignment of two sub-nets towards better learning generalization. Such proof can be easily adapted to more than two sub-nets. The detailed explanation of Eq. 4 is relegated into the Appendix.

### 3.2 Proposed AST with Gradient Correction on Consecutive Inner-group Sub-nets (AST-GC)

Although the implicit regularization of mini-batch SGD helps to maximize the inner product gradient of consecutive sub-nets as discussed in Section 3.1, we indeed observe that there still remain partial number of negative inner product during training, causing conflicting direction of weight updates among sub-nets as shown in Figure 4. However, in terms of optimizing a network by mini-batch SGD, the noise caused by such conflicting gradient direction helps to escape from saddle

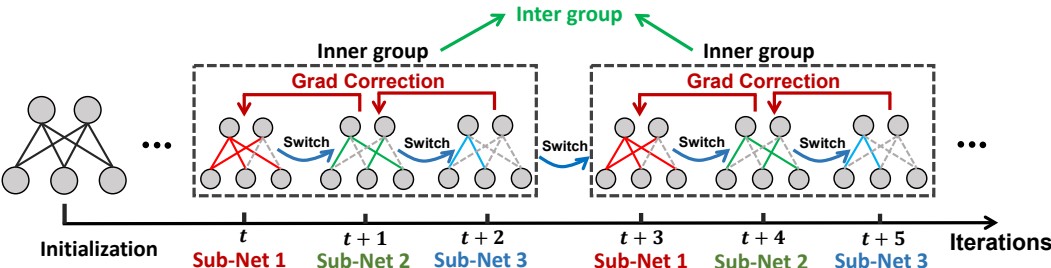

Figure 3: The overview of the AST process with gradient correction on consecutive inner-group sub-nets (AST-GC)

.

points or local minima and improve the generalization as well [6, 1]. Due to that, inspired by the gradient projection method, which is used in multi-task learning [25] and continual learning [12], we propose a *gradient correction* technique to remove the conflicting gradients within the inner-group iterations during training, meanwhile allow the gradient to have negative direction for the inter-group sub-nets. As shown in Figure 3, the *inner-group iterations* is defined as the consecutive sub-nets within $N$ mini-bath iterations, where $N$ is the total number of sub-nets (e.g., $N$=3 as shown in Figure 3). In contrast, inter-group represents the relationship between inner-group iterations. Specifically, the gradient correction adapts a simple procedure within the inner-group sub-nets: if the gradients between two consecutive sub-nets are in conflict (i.e., their cosine similarity is negative), we project the gradient of current sub-net onto the normal plane of the gradient of its prior sub-net. Otherwise, the original gradient is unchanged. Considering two sub-nets in inner-group iterations with the gradient $g_i$ and $g_j$ respectively, the technique can be mathematically illustrated as:

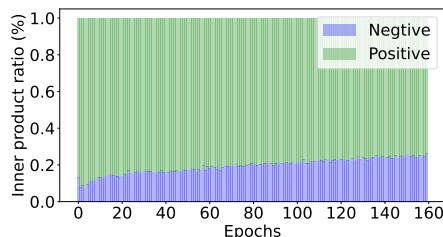

Figure 4: The ratio of negative and positive inner product of two sub-nets during the AST process on CIFAR-10 by using wide ResNet-32.

$$g_i = \begin{cases} g_i - \frac{g_i \cdot g_j}{||g_j||^2} g_j, & \text{if } g_i \cdot g_j < 0 \\ g_i, & \text{otherwise} \end{cases} \tag{5}$$

By doing so, the proposed gradient correction within inner-group iterations have two benefits: 1) it could guarantee that all sub-nets within the consecutive inner-group are updated in the non-conflicting direction, reducing the weight interference between each other; 2) the normal update using the mini-batch SGD between inter-group sub-nets will help to improve the generalization of the model.

### 3.3 Sparse Sub-net Training

Another important aspect of our AST scheme is to generate and train "*sparse*" sub-nets. Following most sparse training works, we adopt the *prune-and-regrow* scheme from GraNet [14] as the backbone technique. Specifically, AST starts from a random initialized sparse model, and then applies prune-and-regrow mechanism as described in Eq. 1 and Eq. 2 for each training iteration with the sub-net specific sparsity ratio. The pseudocode of the proposed AST algorithm is relegated into the Appendix. Given to the fact that alternating training scheme switches the sub-net per iteration, the following questions arises: *1) What is the optimal architecture relationship between sub-nets? 2) How the pruning should be scheduled for each sub-net?* We address these two questions with the following observations:

**Observation 1:** *Enabling the freedom of exploring unique architectures of each sub-net elevates the learning capacity of AST over the S-Net.*

To validate this observation, we generalize the sub-net relations into the following two categories:

- **Completely subset (CS):** As proposed by S-Net [24], the high-sparsity models are fully contained in the low-sparsity models (Figure 5(a)). Under the context of prune-and-regrow, the regrowing process is only performed in the lowest sparsity model, while the rest of

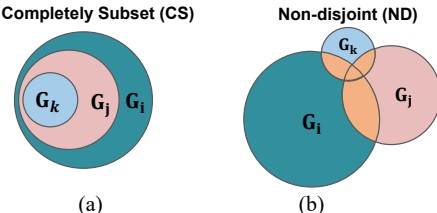

(a)      (b)

Figure 5: Different relationships between three subnets $G_i$, $G_j$, and $G_k$: Completely subset (CS) and Non-disjoint (ND).

Table 1: Averaged accuracy and standard deviations between different AST training schemes on CIFAR-10 dataset (three times experiments each).

| Dataset | CIFAR-10 | | |
|---|---|---|---|
| ResNet-32 | Dense Model Acc. = 94.88 | | |
| Pruning Ratio | 70% | 95% | 98% |
| Completely Subset (CS) | $93.69 \pm 0.12$ | $93.43 \pm 0.03$ | $92.49 \pm 0.17$ |
| **Non Disjoint (ND)** | $\mathbf{94.47 \pm 0.10}$ | $\mathbf{93.78 \pm 0.09}$ | $\mathbf{92.76 \pm 0.21}$ |

the sub-nets rigorously extend the sparsity from the previous low-sparsity model by using magnitude-based pruning only. The one-time regrow guarantees the fully-subset relationships among different sub-nets.

- **Non disjoint (ND):** Each sparse sub-net performs prune-and-regrow individually to optimize the overall pruning decision during training. As depicted in Figure 5(b), sub-nets can freely exploit their own architectures while the intersections remain non-empty. Compared to the CS scheme, the non-disjoint AST empowers the subset networks with more architecture freedom. Figure 6 shows the layer-wise sparsity gap and non-overlap between two ND-trained sub-nets that target different final sparsity values ($s_f$). The non-overlap is computed by XORing the binary sparse masks between sub-nets, the percentages of "1" in the resultant tensor represents the level of non-overlap. Apparently, the distinction of connections is larger than the sparsity difference, which implies the existence of the unique connections generated by the ND prune-and-regrow in each sub-net.

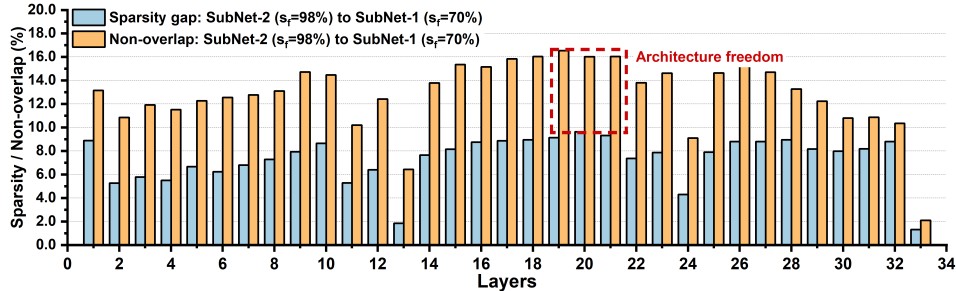

Figure 6: Layer-wise sparsity and the connection dissimilarity between two ResNet-32 sub-nets trained by the non-disjoint (ND) AST scheme.

With the ND scheme, we observe that the small amount of architecture freedom shown in Figure 6 can lead to the improved overall performance: Table 1 summarizes the accuracy of the wide ResNet-32 [26] trained by different AST schemes on CIFAR-10 dataset. Assisted by the comprehensive architecture exploration, the non-disjoint AST scheme achieves higher accuracy compared to the conservative completely-subset (CS) training. Thus, in this work, we use ND-scheme for all experiments.

**Observation 2:** *Intermittent sparsity increment among sub-nets stabilizes AST process.* As introduced in Section 2.1, the sparsity of the in-training sparsification method is periodically updated (e.g., 1,000 iterations) based on a predefined sparsity schedule [14]. Regarding the AST scheme, the sub-net model architectures are consecutively switched and trained, but the successive architecture switching does not imply the necessity of consecutive sparsity update of each sub-net. On the contrary, the intensive sparsity increment of all sub-nets could destabilize the training. In this work, the sparsity of each sub-net increases periodically, but the spar-

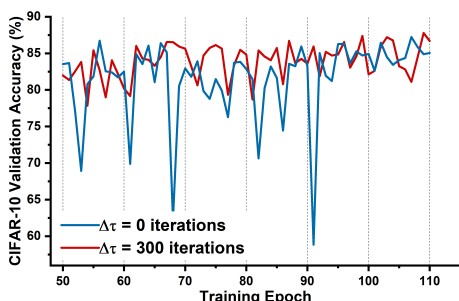

Figure 7: Stabilized training of 98% subnetwork rewarded from the extended adjustment period $\Delta\tau$.

Table 2: CIFAR-10/100 accuracy and training cost comparison with SoTA works on wide ResNet-32.

| Dataset | CIFAR-10 Acc. (%) | | | CIFAR-100 Acc. (%) | | | Train. Cost |
|---|---|---|---|---|---|---|---|
| **ResNet-32** | Dense Model Acc. = 94.88 | | | Dense Model Acc. = 74.94 | | | |
| **Pruning Ratio** | 90% | 95% | 98% | 90% | 95% | 98% | |
| **Individual Training** | | | | | | | |
| Lottery Ticket [5] | 92.31 | 91.06 | 88.78 | 68.99 | 65.02 | 57.37 | 3× |
| SNIP [11] | 92.59 | 91.01 | 87.51 | 68.89 | 65.22 | 54.81 | 3× |
| DSR [17] | 92.97 | 91.61 | 88.46 | 69.63 | 68.20 | 61.24 | 3× |
| GraNet [14]($s_i = 0\%$) | 94.12 | 94.02 | 92.98 | 73.18 | 72.56 | 69.89 | 3× |
| MEST [26]($s_i = 90\%$) | $92.12 \pm 0.13$ | $90.86 \pm 0.11$ | $88.78 \pm 0.26$ | $69.35 \pm 0.36$ | $67.85 \pm 0.23$ | $62.58 \pm 0.31$ | 3× |
| MEST+EM[26]($s_i = 90\%$) | $92.56 \pm 0.07$ | $91.15 \pm 0.29$ | $89.22 \pm 0.11$ | $70.44 \pm 0.26$ | $68.43 \pm 0.32$ | $64.59 \pm 0.27$ | 3× |
| MEST+EMS[26]($s_i = 90\%$) | $93.27 \pm 0.14$ | $92.44 \pm 0.13$ | $90.51 \pm 0.11$ | $71.30 \pm 0.31$ | $70.36 \pm 0.05$ | $67.16 \pm 0.25$ | 3× |
| **Training once for all** | | | | | | | |
| Jointly-Trained [24]($s_i = 0\%$) | $92.59 \pm 0.21$ | $92.58 \pm 0.25$ | $92.48 \pm 0.24$ | $70.40 \pm 0.14$ | $69.32 \pm 0.84$ | $66.85 \pm 0.59$ | 3× |
| AST($s_i = 0\%$) | $93.51 \pm 0.06$ | $93.44 \pm 0.08$ | $92.44 \pm 0.04$ | $73.12 \pm 0.10$ | $72.39 \pm 0.14$ | $68.06 \pm 0.21$ | 1× |
| AST($s_i = 90\%$) | $92.32 \pm 0.06$ | $92.19 \pm 0.11$ | $91.34 \pm 0.04$ | $69.82 \pm 0.12$ | $69.22 \pm 0.07$ | $66.37 \pm 0.15$ | 1× |
| **AST+GC**($s_i = 0\%$) | $\mathbf{93.88 \pm 0.19}$ | $\mathbf{93.70 \pm 0.28}$ | $\mathbf{92.69 \pm 0.09}$ | $\mathbf{73.41 \pm 0.04}$ | $\mathbf{72.57 \pm 0.15}$ | $\mathbf{68.42 \pm 0.15}$ | 1× |
| **AST+GC**($s_i = 90\%$) | $\mathbf{92.90 \pm 0.13}$ | $\mathbf{92.88 \pm 0.10}$ | $\mathbf{91.97 \pm 0.18}$ | $\mathbf{70.11 \pm 0.39}$ | $\mathbf{70.01 \pm 0.54}$ | $\mathbf{67.15 \pm 0.31}$ | 1× |

sity increment of each sub-net is intermittently performed with the adjustment period $\Delta\tau$. In the meantime, sub-nets are still consecutively switched during the $\Delta\tau$. Figure 7 shows the CIFAR-10 validation accuracy of an AST-trained ResNet-32 sub-net with final sparsity of 98%. During the early stage of training with high learning rate, the stabilized validation accuracy indicates the effectiveness of the extended adjustment.

## 4 Experiments

### 4.1 Settings

The proposed AST method is thoroughly verified with multiple datasets, including CIFAR-10, CIFAR-100, and ImageNet. We use similar training scheme as prior works [14, 26], which employed 160 epochs for CIFAR-10/100 and 100 epochs for ImageNet training. The multiple sparse sub-nets are trained from scratch and pruned by the proposed AST algorithm. The cosine annealing learning rate schedule is employed with 0.1 initial learning rate for the CIFAR datasets training. For ImageNet dataset, we introduce extra 5 epochs for the initial warmup training. The pruning candidates are globally selected while the first layer remains dense. Regarding the regrowing process, the percentage of regrow candidates gradually decrease from 0.5 to 0.0 with cosine annealing schedule. In the meantime, the extended adjustment period between sub-nets is set to 100 and 400 for CIFAR and ImageNet experiments, respectively. While we believe a more fine-grained hyperparameter tuning could lead to better accuracy, we choose the above scheme for simplicity and reproducibility. The reported sub-net accuracy values are obtained from a single background model checkpoint. In all experiments, we report the average accuracy with its variation in 3 runs.

### 4.2 Main Results

**CIFAR-10 and CIFAR-100.** Table 2 shows the CIFAR-10/100 accuracy of the proposed AST algorithm, where we used the wide ResNet-32 model that was employed in [26]. Following the typical high sparsity results reported previously, we train three sub-nets with 90%, 95%, and 98% sparsity from scratch all at once with identical epochs as prior works. Given the fact that the high initial sparsity is beneficial to the overall memory saving of the training process, we report the results with both dense ($s_i = 0\%$) and highly sparse ($s_i = 90\%$) initial models. Compared to the prior individually-trained SoTA works [14, 26], the proposed AST+GC algorithm obtains three highly sparse models through one-time training with negligible accuracy degradation. The resultant 3× total training cost reduction frees the model from energy-urged extra training, leading to power and latency benefits. The joint training scheme [24] requires multiple forward pass in each iteration, the increased computational cost leading to the averaged-yet-suboptimal performance. In addition to the reduced training cost, the proposed AST method outperforms the joint-training scheme [24] by up to 1.3% and 2.9% inference accuracy, while achieving up to 2.63× training cost reduction on CIFAR-10 and CIFAR-100 datasets, respectively. We also verify the AST scheme with more sub-nets and large sparsity gaps in Section 4.4.

**ImageNet-2012.** We further evaluate the proposed method with ResNet-50 on ImageNet in Table 3. It is noticeable that the 0.5× and 0.25× model of the jointly-trained S-Net [24] corresponding to

Table 3: ImageNet accuracy and training cost comparison with SoTA works on ResNet-50.

| Method | ImageNet-2012 | | | |
|---|---|---|---|---|
| **ResNet-50** | Dense model Acc. = 76.8 | | | |
| **Prune Ratio** | 80% | | 90% | |
| | **Individual Training** | | | |
| | Top-1 Acc. (%) | Training Cost | Top-1 Acc. (%) | Training Cost ($\times$e18) |
| SNIP [11] | 69.7 | 1.67 | 62.0 | 0.91 |
| SET [15] | 72.9 | 0.74 | 69.6 | 0.32 |
| DSR [17] | 73.3 | 1.28 | 71.6 | 0.96 |
| RigL [4] | 74.6 | 0.74 | 72.0 | 0.39 |
| MEST + EM [26] | 75.8 | 0.74 | 73.6 | 0.39 |
| GraNet [14] | 76.0 | 1.18 | 74.5 | 0.80 |
| | **Training once for all** | | | |
| Jointly-Trained [24]($s_i = 50\%$) | $71.9_{0.5\times}$ | 1.19 | $65.0_{0.25\times}$ | 1.19 |
| **AST ($s_i = 50\%$)** | **72.6** | **0.59** | **72.3** | **0.41** |
| **AST + GC ($s_i = 50\%$)** | **73.2** | **0.59** | **73.1** | **0.41** |
| **AST + GC ($s_i = 80\%$)** | **72.6** | **0.37** | **72.5** | **0.13** |

Table 4: Inference acceleration and negligible accuracy drop of the proposed AST algorithm with structured fine-grained sparsity on ResNet-18 model.

| Dataset | CIFAR-10 Acc. (%) | | | | | Training Cost |
|---|---|---|---|---|---|---|
| N:M Sparse Pattern | Dense Model | 2:4 | 3:4 | 7:8 | 15:16 | |
| Individually Trained (SR-STE) [27] | 95.07 | 94.89 | 94.47 | 94.25 | 93.92 | 2.33e+16 (3.95$\times$) |
| **AST + GC** | - | 94.63 | 94.26 | 94.31 | 93.79 | **5.91e+15 (1$\times$)** |
| **Inference Time / 10K images (sec)** | 1.40 | 1.01 | 0.67 | 0.66 | 0.63 | - |

72.98% and 92.23% weight sparsity, the averaged overall sparsity (82.6%) is less than our targeted sparsity (85%). The proposed AST training scheme outperforms the joint-training scheme by 7.5% inference accuracy with 80% highly sparse initial models. Compared to the individual training scheme, the proposed AST scheme achieves a comparable or even better performance to SNIP [11] and SET [15], while reduces the total training cost up to 2.38$\times$. In addition, compared to AST without gradient correction (GC), AST+GC improves accuracy by 0.6% and 0.8% respectively. The experimental results with different settings and models are shown in the Appendix.

### 4.3 Extend to Structured Fine-grained Sparsity

Demonstrating the sparsity-induced speedup on GPU has now emerged as a feasible solution to depict the effectiveness of pruning algorithms. The recent Nvidia Ampere architecture is equipped with the *Sparse Tensor Cores* to accelerate the neural network computation on GPU with $N{:}M$ structured fine-grained sparsity [27]. Varying the group size $M$ and the number of sparse elements $N$ leads to different overall sparsity values. Motivated by this, we extend the proposed AST algorithm to train multiple sub-nets with different $N{:}M$ sparsity patterns all at once. Specifically, the prune-and-regrow is performed based on the group-wise summed weight or gradient magnitude. Powered by the open-sourced Nvidia-ASP library, the convolution computation can be accelerated when the sparse weight group size $M$ is divisible by 4 (e.g., 4, 8, 16). The proposed AST algorithm collectively trains four ResNet-18 sub-nets with 2:4, 3:4, 7:8, and 15:16 sparse patterns, corresponding to 50%, 75%, 87.5%, and 93.75% overall sparsity, respectively. Starting from scratch, the percentage of the $N{:}M$ sparse groups is gradually increased from 0% to 100% within each sub-net. Compared to the individually trained dense model, the proposed AST scheme achieving up to 2.3$\times$ inference speed up on GPU with 4$\times$ less training efforts and negligible accuracy loss, as shown in Table 14. The inference time is measured on a Nvidia 3090 GPU with FP32 data precision.

### 4.4 Ablation Study

**AST with more than 3 sub-nets.** We also verify the effectiveness of the proposed AST scheme for training different number of subset networks. Table 5 shows the CIFAR-10 performance of AST by training 2, 3, 4, 5 sparse sub-networks collectively based on ResNet-18. Compared to the individually-trained baseline sparse model, training more sub-nets with AST reduces the total training effort with the cost of the marginally degraded accuracy, especially in the high sparsity models (e.g., 98% sparsity). Given the tradeoffs between accuracy and total training cost, training three sub-nets have the best overall performance. The flexibility of training a large number of sub-nets all at once enables the proposed AST algorithm to select suitable architecture for different power budgets.

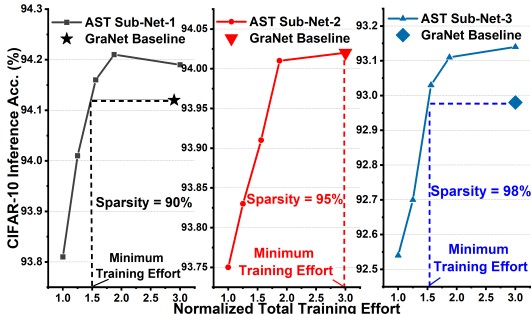

Figure 8: AST with extended training effort on CIFAR-10 with wide ResNet-32 [26].

Table 5: ResNet-18 training results of AST (160 epochs) with various sparsity values and numbers of sub-nets for CIFAR-10 dataset.

| Dataset | CIFAR-10 Acc. (%) | | | | |
|---|---|---|---|---|---|
| Sparsity | Indiv. trained | 2 sub-nets | 3 sub-nets | 4 sub-nets | 5 sub-nets |
| 70% | 95.11 | 94.88 | 94.75 | 94.79 | 94.81 |
| 80% | 94.94 | - | - | 94.81 | 94.73 |
| 90% | 94.93 | - | - | - | 94.85 |
| 95% | 94.88 | - | 94.67 | 94.63 | 94.47 |
| 98% | 94.50 | 94.76 | 94.26 | 94.22 | 94.18 |

Table 6: ImageNet accuracy with different sparsity combinations on ResNet-50.

| Method | ImageNet-2012 | | | | | |
|---|---|---|---|---|---|---|
| ResNet-50 | Dense Model Acc. = 76.8 | | | | | |
| Prune Ratio | 50% | 90% | 50% | 95% | 80% | 90% |
| AST+GC (s = 0%) | 74.68 | 73.26 | - | - | - | - |
| AST+GC (s = 0%) | - | - | 74.21 | 71.07 | - | - |
| AST+GC (s = 50%) | - | - | - | - | 73.2 | 73.1 |
| AST+GC (s = 80%) | - | - | - | - | 72.6 | 72.5 |

**AST sub-nets with different sparsity difference.** Besides the collective training of highly sparse models, learning the largely-varied subset architectures is also essential. As shown in Table 5 and 6, the proposed AST algorithm is still able to optimize the model performance with the large sparsity gaps (e.g., 50% vs. 95%).

**Extended AST training efforts.** As shown in Figure 2, AST iteratively selects different sub-net for each mini-batch iteration. Even though the batch shuffling is employed to the training, each sub-net cannot be fully trained by the whole training set inside each epoch. A straight-forward solution is extending the total training efforts (epochs). Assume the unit training cost is 160 epochs ($1\times$), we evaluate the AST performance with three wide ResNet-32 [26] sub-nets with different training effort, as shown in Figure 8. Compared to the individually trained GraNet [14] baseline (total=$3\times$), AST achieves the same accuracy in all three sparse models with only $\sim 2\times$ averaged total training effort.

## 5 Conclusion

In this work, we first propose Alternating Sparse Training (AST) scheme to train multiple sparse neural networks all at once. Then, we demonstrate the benefits of gradient correction (GC) with theoretical analysis and experimental results. As one of the earliest research works in this domain, the AST algorithm achieves high accuracy and high training efficiency without any repeated or ensembled training steps [24]. Inspired by the prune-and-regrow scheme [14], the proposed AST-GC scheme exploits multiple sparse sub-nets simultaneously while achieving comparable or even higher accuracy against the individually-trained SoTA methods. Obtaining multiple models with AST brings practical benefits to the energy-driven hardware computation along with superior performance on accuracy.

**Impact and limitation.** In the current work, AST can work well on a certain number of sub-nets. Extending it to support a larger number of sub-nets is an interesting direction. In addition, to enlarge the practical efficiency for both training and dynamic inference, we will further apply quantization and investigate more structured sparsity pattern upon AST. Currently, we have demonstrated the practical value for dynamic inference on GPU with Nvidia Ampere architecture by using N:M structured sparsity pattern. We would like to further demonstrate the actual efficiency of AST on more hardware platforms, such as CPU, mobile/IoT devices, specific hardware accelerators, etc.

**Acknowledgments.** We thank the anonymous reviewers for their constructive comments. This work is supported in part by the National Science Foundation under Grant No.1931871, No.2144751, No.1652866, No.1931871, No.2144751, and the Center for Brain-inspired Computing (C-BRIC) in JUMP, an SRC program sponsored by DARPA.

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
