# OpenReview forum: "Get More at Once: Alternating Sparse Training with Gradient Correction"
_NeurIPS.cc/2022/Conference — NeurIPS 2022 Accept_

### Official Review · Reviewer_sm86 · 2022-07-05

**Rating:** 5
**Confidence:** 5
**Soundness:** 3 good
**Presentation:** 3 good
**Contribution:** 1 poor

**Summary:**

This paper proposes to alternatively train multiple sparse sub-nets to obtain a supernet that supports dynamic inference with varying sparse rates, termed AST. Experiments on ImageNet and CIFAR-10/100 validated the efficacy of the proposed AST, which is also characterized by efficient training compared with traditional supernet methods.


**Questions:**

1. For the gradient correction part, does Eq.5 work if the overlap of two subnets is very small?
2. I recommend the authors provide experiments of structured sparsity to demonstrate the efficiency of AST against S-Net, US-Net, OFA.

**Ethics Review Area:**

["I don’t know"]

**Limitations:**

The authors do not discuss the limitations of their work as well as the negative societal impact.

**Strengths And Weaknesses:**

Strengths:
1. The proposed grad correction mechanism is well-motivated.
2. The manuscript is well-organized and the proposed AST is easy to follow.

Weaknesses:
1. The practical value of AST is highly limited. 1) It is a consensus that unstructured pruning methods hardly gain acceleration on general hardware. In contrast, the advantage of unstructured pruning falls into significant parameter saving (storage reduction) without performance compromise. However, AST trains multiple subnets towards "dynamic inference" for sparse networks, which is unreasonable since dynamically running sparse subnets can not save inference costs as structured methods do. Moreover, the resulted supernets even bring more parameter burden than classic sparse training methods. (Although the authors have provided N:M fine-grained experiments, the patterns are selected with 2:4, 3:4, 7:8, 15:16. Here 3:4, 7:8, 15:16 patterns can not be applied on real hardware as far as I can know. Also, the experiments are only conducted on CIFAR-10. Overall, it is not convincing to me.)
2. Lack novelty. The key component of AST is combining prune-and-regrow and supernet-training. Despite the originality of gradient correction, the limited practical value also degrades the overall contributions of this paper.
3. The performance of AST is poor, compared with SOTA sparse training methods like GraNet.

---

> ### Author Response · Authors · 2022-08-02
> **Response to Reviewer sm86**
>
> **Q1**: The practical value of AST is highly limited.
>
> **A1**: We respectfully disagree with the reviewer's opinion. First, for the first comment regarding to the unstructured pattern, if the difference is relatively small (for iso-accuracy, if structured sparsity value is lower than the unstructured sparsity by a small amount), then hardware acceleration would be better with structured sparsity.  However, the sparsity of the proposed unstructured sparsity scheme in this work is reaching 95\% and 98\%, so only 5\% and 2\% of the total weights need to be stored with corresponding indexes.
>
> Second, to demonstrate the practical value of the proposed AST, we have measured the actual inference speedup on all the selected N:M sparsity (2:4, 3:4, 7:8, 15:16) patterns by using the Nvidia-ASP library [1] on GPU as shown in Table 4 of the original
> manuscript. Moreover, the actual inference speedup of the selected N:M sparsity have been demonstrated in prior work [2] as well, where the group size M varies from 4 to 16.
>
> We would like to claim that it is not true that 3:4, 7:8, and 15:16 patterns cannot be applied on real hardware, as the reviewer mentioned.
>
> Third, the reviewer omits the advantage of the dynamic inference ability of the proposed AST. AST could enable run-time dynamics and allows users to tune the tradeoffs between model running complexity and inference accuracy in real-time.
>
> [1] Zhang, Jie-Fang, et al. "Snap: An efficient sparse neural acceleration processor for unstructured sparse deep neural network inference." IEEE Journal of Solid-State Circuits 56.2 (2020): 636-647.
>
> [2] https://github.com/NVIDIA/apex
>
> [3] Aojun Zhou, et al. Learning n: m fine-grained structured sparse neural networks from scratch. arXiv preprint arXiv:2102.04010, 2021.
>
> **Q2**:  Lack novelty.
>
> **A2**: We kindly disagree the statement regarding the ``lack of novelty''. To appreciate the prior works, we discussed and cited prune-and-regrow and supernet-training works. We want to clarify that our AST methodology adopted these two prior methods, but we did not claim our novelties come from the combination of these two methods.  We would like to highlight our novelties contributions as below:
>
> 1)  To reduce the training cost in prior supernet training for dynamic inference, we propose an alternating sparse training method (AST), which is highly distinguished from prior dynamic inference methods (i.e., S-NN, US-NN, OFA, etc.).  Specifically, our AST generates multiple sparse sub-nets at one training run without extra training cost by alternately training sub-nets with different sparsity values (please also refer to the response to Reviewer x1xA).
>
> 2) Based on our finding of conflicting gradient issues between sub-nets in AST, we propose a gradient correction to remove the conflicting gradient direction between sub-nets. Our AST achieves the best SOTA accuracy and training cost performance for training multiple sub-nets based dynamic network all-at-once.
>
> **Q3**: The performance of AST is poor, compared with SOTA sparse training methods like GraNet.
>
> **A3**: We respectfully disagree with the comment on the poor performance. Instead, we want to emphasize that our proposed AST achieves the best SOTA accuracy and training cost performance for training multiple sub-nets based dynamic network all-at-once. In the experiments, we compare the proposed AST with individual-training methods and jointly-training methods, respectively. Jointly-training method (i.e., S-NN) is the baseline of the proposed AST since it can enable dynamic inference. Compared to S-NN, we have shown that AST significantly improves the accuracy of both CIFAR and ImageNet datasets with much less training cost, achieving SOTA.
>
> In contrast, individual-training methods, like GraNet, aim to train a single sparse model by a prune-and-regrow scheme, which has no ability to make dynamic inference and could be considered as the upper-bound of the joint-trained method.
> We provide comparison to GraNet to just show that our multi-sub-net training scheme could achieve the accuracy close to one single sparse network %training accuracy with the same training cost, while providing a group of sub-nets with different sparsity ratios.
>
> **Q4**: Additional Question.
>
> **A4**: For the comment of Q4-1, we think the gradient correction will have little effect if the overlap of two sub-nets is very small. However, we observe that subnets are highly shared between each other (i.e., more than 90\% of their weight), which promotes the utilization of the proposed gradient correction. Furthermore, such property is beneficial to the practical hardware deployment for dynamic inference since the extra parameter storage is negligible compared to a single sparse model (refer to Q1).
>
> For the comment of Q4-1,  due to the limited rebuttal time, we couldn't obtain the detailed structured sparsity results yet. But we are investigating this and will include them in the next revised version.

---

> > ### Comment · Reviewer_sm86 · 2022-08-08
> > **Response**
> >
> > Thank you for the reply, which partially addressed my concerns.
> >
> > Q1: The NVIDIA ASP library only supports 2:4 pattern as far as I know. In my view, patterns like 3:4/15:16 are indeed not hardware-friendly.
> >
> > Q4: I see, but an ablation studies to show the overlap would be better for clarification.

---

> > > ### Author Response · Authors · 2022-08-09
> > > **Reply to Reviewer sm86**
> > >
> > > Thank you for your further comments. We have the following clarifications:
> > >
> > > **Q1**: The NVIDIA ASP library only supports 2:4 pattern as far as I know. In my view, patterns like 3:4/15:16 are indeed not hardware-friendly.
> > >
> > > **A1**: We respectfully disagree with this comment. In the experiments, we adapt the implementation from the NVIDIA ASP official codebase [[link]](https://github.com/NVIDIA/apex/blob/cd0a1f11061068db45f12ef829ca3250389cd7ae/apex/contrib/sparsity/sparse_masklib.py#L50) by changing the settings from 2:4 to different N:M groups accordingly (i.e., 3:4, 7:8, 15:16).
> > > We want to emphasize again that **we measured the actual inference speedup on the select sparsity pattern.** For example, as shown in the Table.4 of the original manuscript, 3:4 and 15:16 sparsity patterns achieve **2x and 2.3x inference speedup** compared to the corresponding dense model respectively. Such results demonstrate that the NVIDIA ASP library does support the selected sparsity pattern although the sparsity pattern 15:16 does not support well as 2:4. We will release our code on Github if paper is accepted to let any reader reproduce our experiment results.
> > >
> > > **Q4**: An ablation study to show the overlap would be better for clarification
> > >
> > > **A4**: We thank the reviewer for your suggestion. We add the ablation study about the overlap between sub-nets on CIFAR-10 and CIFAR-100 respectively as shown in Table.R1 below. Note that, 'Overlap: 95\% to 90\%' means the overlap ratio of the weights between the 95\% and 90\% sparse sub-nets. Similarly, 'Overlap: 98\% to 90\%' means the overlap ratio between the 98\% and 90\% sparse sub-nets. **It is clear to see that the weights are highly shared between different sub-nets in both settings and datasets.**
> > >
> > > Table R1: Sub-nets overlap analysis on CIFAR-10 and CIFAR-100 by using wide ResNet-32.
> > > |  Dataset  | Overlap: 95% subnet to 90% subnet | Overlap: 98% subnet to 90% subnet |
> > > | :-------: | :-------------------------------: | :-------------------------------: |
> > > | CIFAR-10  |              92.62%               |              87.87%               |
> > > | CIFAR-100 |              93.58%               |              89.73%               |
> > >
> > >
> > > Thanks again for all your response and fruitful comments. We hope our detailed response can resolve your questions and concerns, please kindly reconsider the overall score of our work.

---

> > > > ### Comment · Reviewer_sm86 · 2022-08-09
> > > > **Response**
> > > >
> > > > Thanks for your quick reply, I will raise my score to 5 and look forward to the code for actual inference speedup on 3:4/15:16 patterns. Besides, N:M sparsity generally considers N as non-zero elements in M weights block. So the claimed 3:4/15:16 patterns are confusing for the readers (That is why I think 15:16 is not hardware-friendly before). I recommend the authors make a unified expression. Please refer to:
> > > >
> > > > 1. Learning N:M Fine-grained Structured Sparse Neural Networks From Scratch. In ICLR, 2021.
> > > > 2. Accelerated sparse neural training: A provable and efficient method to find n: m transposable masks. In NeurIPS, 2021.

---

> > > > > ### Author Response · Authors · 2022-08-09
> > > > > **Many thanks for your further updates!**
> > > > >
> > > > > We thank the reviewer very much for further reviewing our response and increasing the score. We will adapt the same expression of N:M pattern as the references above in the next revision.

---

> ### Author Response · Authors · 2022-08-08
> **Look forward to your feedback**
>
> Dear reviewer sm86,
>
> Thanks again for your time. As the deadline for discussion is approaching, we really hope to have a further discussion with you to see if our response solves the concerns. We are happy to provide any additional clarifications that you may need.
>
> Best wishes, Author

---

### Official Review · Reviewer_MN8L · 2022-07-09

**Rating:** 5
**Confidence:** 4
**Soundness:** 3 good
**Presentation:** 2 fair
**Contribution:** 2 fair

**Summary:**

This paper introduces an algorithm for obtaining multiple sparse neural networks with different sparsity levels in a single training run. The authors also propose a gradient correction strategy that contains projecting the gradient of a sub-net to the plane of the prior sub-net; the authors explain that this makes the direction of the gradient updates among various models consistent.

**Questions:**

Most of the concerns/suggestions have been discussed in the previous section. I add some minor comments here:
- In some places, descriptions of methods/experiments are not clear. in Section 3 (or 2), I suggest authors add a formal definition of a sub-net. At first read, it is not clear how W_i are related, and they are coming from the same model. Adding a pseudo-code can help to better clarify this.
- Typos: Lines 139 (,),233 (remains), 290 (iteratively)
- Line 260, The results from the Appendix are better to be included in the paper.
- Figure 8. It’s better to include the sparsity levels for each sub-network in the figure.
- How many runs are performed for each experiment?
- I believe [1] is a related work in terms of deriving multiple sparse networks at a single run.

[1] Liu, S., Chen, T., Atashgahi, Z., Chen, X., Sokar, G., Mocanu, E., ... & Mocanu, D. C. (2021). Deep ensembling with no overhead for either training or testing: The all-round blessings of dynamic sparsity. arXiv preprint arXiv:2106.14568.


**Limitations:**

Limitations are not discussed in the paper.

**Strengths And Weaknesses:**

Strengths:

- The proposed method is efficient in terms of deriving multiple sparse networks during a single round of training.
- Figures 2 and 3 describe the proposed methods clearly.
- While the writing and the descriptions can be improved, the paper is well-organized.

Weaknesses:

- One of the main concerns is that while the method derives multiple sub-networks, the performance of each sub-network falls behind the competitors in several cases. The reported accuracies are mostly compared to a single method [23]. I think the results should be compared with more methods given similar training costs to evaluate the quality of the sub-network. E.g., authors can consider deriving subnetworks with similar sparsity levels during a single run of GraNet and compare the accuracies.
- The training costs are given in terms of training rounds. However, as different methods can have different training costs, the costs should be reported using another metric, e.g., FLOPs. Memory costs of keeping the masks for each sub-network need to be discussed more clearly.
- AST-GC requires extra computation at Equation 5, which needs to be considered in the discussions and FLOPs computation.
- Some points are referenced in the Appendix; however, they are missing. E.g., Line 147 regarding discussion on Equation 4. Line 388 Limitations of the work
While the general idea is interesting, I believe the paper is not ready for publication yet.

---

> ### Author Response · Authors · 2022-08-02
> **Response to Reviewer MN8L**
>
> **Q1**: One of the main concerns is that while the method derives multiple sub-networks, the performance of each sub-network falls behind the competitors in several cases.
>
> **A1**: Thank you for your suggestion.
> First, we want to emphasize that we already achieve the best SOTA accuracy and training cost performance for training multiple sub-nets based dynamic network all-at-once. The comparison with other related run-time dynamic network are in Table 2 and Table 3 of the original manuscript for CIFAR and ImageNet respectively. We provide comparison to GraNet to just show our multi-sub-net training scheme could achieve close to one single sparse network training accuracy with the same training cost, but providing a group of sub-nets with different sparsity ratios. GraNet is an in-training sparsity method that only generates a single sparse network in one run of training. As suggested by the reviewer, to derive multiple sub-nets using GraNet method, we progressively train three sparse sub-nets with the same total number of training epochs as the proposed AST by using the GraNet method. As presented in Table 10 of the revision, with the wide ResNet-32 model, we first sparsify the model to 90\% sparsity from scratch with 60 epochs. Subsequently, we prune the 90\% model to 95\% and then to 98\% sparsity with consecutive 50 epochs. Note that the total training effort remains the same (60+50+50=160 epochs) as a single training run of AST. The experimental results show that such a progressive tuning method failed to achieve similar performance as the proposed AST, as shown in Table 11 of the revision.
>
> **Q2**: The training costs are given in terms of training rounds. However, as different methods can have different training costs, the costs should be reported using another metric, e.g., FLOPs. Memory costs of keeping the masks for each sub-network need to be discussed more clearly.
>
> **A2**: Thank you for your response. We updated the training cost in terms of FLOPS and summarized the results in Table 8 in the Appendix of the revision. Compared to the joint-training scheme S-Net or separately trained GraNet, the proposed AST algorithm achieves up to 2.63$\times$ training FLOPS reduction while maintaining the similar inference accuracy as the individual training baseline. The updated Table 3 is included in the Appendix of the revised manuscript.
>
>
> **Q3**: AST-GC requires extra computation at Equation 5, which needs to be considered in the discussions and FLOPs computation.
>
> **A3**: Thank you for pointing this out.  We provide the computation cost comparison in terms of FLOPs, with the consideration of the computing requirement of gradient correction (GC), as summarized in Table 8 of the revison. With the wide ResNet-32 model and CIFAR-100 dataset, training three sparse subset networks with gradient projection requires pair-wise dot product computation inside each inner subnet group (refer to Figure 2 of the original manuscript).  As can be seen, this overhead is minimal compared with the overall training cost of the entire model. For example, the computational FLOPs of gradient correction is 1.856e+12, while the overall training FLOPs of AST (initial sparsity $s_i = 0$) is 6.47e+15.
>
> **Q4**: Some points are referenced in the Appendix.
>
> **A4**: Thank you for pointing this out. (1) The detailed explanation 147 of Eq. 4: we added the description in the Appendix 6.2 of the revised version. (2) For the limitations of the work: we included a paragraph about impact and limitation of proposed AST in the Conclusion section of the revised version.
>
> **Q5**: Minor comments.
>
> **A5**:
> For the comments of Q5-1:  The sub-net is defined as a partial weight of the network, which can run inference independently. All the sub-nets are partially shared with each other within a single model. Due to that, the proposed AST could enable dynamic inference where the sub-nets can be switched at run-time, permitting dynamic inference accuracy and complexity trade-off. In addition, we added the pseudo-code of AST in the Appendix of the revision.
>
> For the comments of Q5-2, Q5-3, and Q5-4, we have addressed the typos and suggestions for line 260 and Figure 8 in the revised version.
>
> For Q5-5, in all experiments, we report the average accuracy with its variation in 3 runs.
>
> For Q5-6, based on the in-training sparsification mechanism, [1] proposes a dynamic sparse training ensemble method to independently generate multiple sparse sub-nets for ensemble. Such method is orthogonal to our work, which could be further combined to improve accuracy.
>
> [1] Liu, S., Chen, T., Atashgahi, Z., Chen, X., Sokar, G., Mocanu, E., ... \& Mocanu, D. C. (2021). Deep ensembling with no overhead for either training or testing: The all-round blessings of dynamic sparsity. arXiv preprint arXiv:2106.14568

---

> > ### Comment · Reviewer_MN8L · 2022-08-08
> > **Response**
> >
> > I thank the author for their response and clarifications.
> >
> > - However, the motivation and the benefits of the algorithm are not fully clear to me yet. My major concern is that what are the benefits of the algorithm if each sub-network has a lower performance than training a single sparse network individually? In Table 3, the performance of the two sub-networks is very close to each other on the ImageNet dataset (0.1% accuracy). So the initial goal to achieve run-time sparsity and accuracy trade-off is not accomplished. What would be the benefits of two networks with different sparsity levels that achieve almost the same accuracy? Why do we not spend our resources on training a single well-performing sparse network with high sparsity?
> >
> > - Will the extended training time (Figure 8, which is done on CIFAR-10) be effective also on the other datasets?
> >
> > - The writing can be significantly improved.

---

> > > ### Author Response · Authors · 2022-08-09
> > > **Reply to Reviewer MN8L**
> > >
> > > Thank you for your further comments. We have the following clarifications:
> > >
> > > **Q1**: The motivation and the benefits of the algorithm are not fully clear to me yet.
> > >
> > > **A1**: We would like to clarify the motivation and benefits of the proposed AST as follows:
> > >
> > > 1) **the proposed AST inherits the motivation and benefits from a series of prior dynamic inference works (e.g., S-Net[1], US-Net[2], OFA[3], BigNas[4], DS-Net[5]), which aim to train a network once to get multiple sub-nets where their weights are highly shared to enable run-time switching for inference (i.e, dynamic inference). Thus, one-time training generated networks could fit for different types of hardware and resource requirements. The motivation and benefits of such dynamic inference are discussed in our introduction part and widely used in our listed prior works below.**
> > >
> > > 2) **compared to prior dynamic inference works, we already achieved the best SOTA accuracy performance, meanwhile largely reducing the training cost.** We believe that our proposed AST makes a significant contribution to the field of dynamic network through significantly improving the training efficiency and accuracy. The small accuracy difference between each sub-net indeed shows the superior performance of our methods since the accuracy difference in prior works are much larger. It is also worth noting that the accuracy difference for different structured sparse sub-nets are much larger as shown in the Table 4 of our original manuscript, where we deployed the N:M structured sparse sub-nets in Nvidia GPU with Ampere architecture to show their accuracy v.s. measured inference time tradeoff.
> > >
> > >
> > > [1] Jiahui Yu, et al. Slimmable neural networks. In International Conference on Learning Representations, 2018
> > >
> > > [2] Yu, Jiahui, and Thomas S. Huang. "Universally slimmable networks and improved training techniques." Proceedings of the IEEE/CVF international conference on computer vision. 2019.
> > >
> > > [3] Cai, Han, et al. "Once-for-all: Train one network and specialize it for efficient deployment." In International Conference on Learning Representations, 2020.
> > >
> > > [4] Yu, Jiahui, et al. "Bignas: Scaling up neural architecture search with big single-stage models." European Conference on Computer Vision. Springer, Cham, 2020.
> > >
> > > [5] Li, Changlin, et al. "Dynamic slimmable network." Proceedings of the IEEE/CVF Conference on Computer Vision and Pattern Recognition. 2021.
> > >
> > > **Q2**: Will the extended training time (Figure 8, which is done on CIFAR-10) be effective also on the236
> > > other datasets?
> > >
> > > **A2**: To further show the effectiveness of the extended training time, we conducted the experiments on CIFAR-100 by using the same setting as CIFAR-10 of the original manuscript. As shown in Table R1 below, extending the total training effort from 160 epochs to 200 and 250 epochs consistently improve the inference accuracy for all three sparse subset models, respectively.
> > >
> > > Table R1: Improved wide ResNet-32 performance of AST with extended training time on CIFAR-100 dataset.
> > > |                Dataset                 |                |     CIFAR-100 Acc (%)     |                |
> > > | :------------------------------------: | :------------: | :-----------------------: | :------------: |
> > > |               ResNet-32                |                | Dense Model Acc. = 74.94% |                |
> > > |            Training Epochs             |      90%       |            95%            |      98%       |
> > > |            160 (1$\times$)             | 73.39$\pm$0.23 |      72.54$\pm$0.14       | 68.58$\pm$0.26 |
> > > |           200 (1.25$\times$)           | 73.46$\pm$0.18 |      72.76$\pm$0.16       | 69.08$\pm$0.11 |
> > > |           250 (1.56$\times$)           | 73.64$\pm$0.14 |      72.94$\pm$0.15       | 69.12$\pm$0.15 |
> > > |          300 (1.875$\times$)           | 74.47$\pm$0.09 |      73.14$\pm$0.23       | 69.97$\pm$0.07 |
> > > | Individually-Trained ($\sim$3$\times$) |     73.18      |           72.56           |     68.56      |
> > >
> > > **Q3**: The writing can be significantly improved.
> > >
> > > **A3**: As we mentioned in our response above. All the reviewer's comments about paper writing have been addressed. We would like to encourage the reviewer to check our submitted revision. We are also glad to further address the reviewer's detailed comments about writing if provided.
> > >
> > > Thanks again for all your response and fruitful comments. We hope our detailed responses can resolve your questions and concerns, please kindly reconsider the overall score of our work.

---

> > > > ### Author Response · Authors · 2022-08-09
> > > > **Hope We Resolve Your Questions and Concerns**
> > > >
> > > > Dear reviewer MN8L,
> > > >
> > > > Thanks again for all your response and fruitful comments. We hope our detailed responses can resolve your questions and concerns, please kindly reconsider the overall score of our work
> > > >
> > > > Best wishes, Author

---

> > > > ### Comment · Reviewer_MN8L · 2022-08-09
> > > > **Response**
> > > >
> > > > Thank you for your clarification. I will increase my score to 5.

---

> ### Author Response · Authors · 2022-08-08
> **Look forward to your feedback**
>
> Dear reviewer MN8L,
>
> Thanks again for your time. As the deadline for discussion is approaching, we really hope to have a further discussion with you to see if our response solves the concerns. We are happy to provide any additional clarifications that you may need.
>
> Best wishes, Author

---

### Official Review · Reviewer_x1xA · 2022-07-10

**Rating:** 6
**Confidence:** 4
**Soundness:** 3 good
**Presentation:** 2 fair
**Contribution:** 3 good

**Summary:**

This paper combines two recently emerging methods, sparse training and dynamic inference, to obtain multiple subnetworks with different sparsity levels. Compared with traditional sparse training methods, the proposed method (AST) is able to obtain various sparse NNs with one training run. Compared with dynamic inference,  AST does not need to perform forward and backward for every sub-net.

I believe this paper, albeit short on technical novelty, does provide a sufficiently novel and important empirical evaluation and details analysis of sparse training and dynamic inference, and could be accepted - assuming the authors can address the issues I have detailed in my review.



**Questions:**

Please refer to the weakness below.

**Limitations:**

yes

**Strengths And Weaknesses:**

## Strengths

(1) To the best of my knowledge, the combination of sparse training and dynamic inference is novel, which can further improve the efficiency of sparse training.

(2) The paper is relatively clear and I can obtain the general idea easily after reading the introduction.

(3) The experimental setting is clear and fair, demonstrating the effectiveness of the proposed methods.

(4) The empirical study in this paper is extensive (e.g., Completely subset (CS) vs Non-disjoint (ND) ), providing insights for researchers who want to follow this topic.

## Weakness

(1) Several naive baselines are missing. This paper only compares with several sparse training baselines and Jointly-Trained, while some naive baselines are missing. On the top of my head, for instance,  (1) train a dense model, and prune to various sparsity with a short time of fine-tuning. (2) train a sparse model with sparse training methods (e.g., GraNet), and prune to various sparsity with a short time of fine-tuning. While this procedure involves fine-tuning, we can still keep the overall training time as same as AST by decreasing the pretraining time.

(2) The accuracy of AST are relatively lower compared with the backbone GraNet on ImageNet. I can imagine that the performance of AST is lower than GraNet, since training one network is intuitively easier than training multiple networks. Is the performance degradation caused by the insufficient updates of each subnet? Since the author keeps the overall training time the same as GraNet, the overall training time of each is 1/N (N is the number of subnets) of the GraNet?

(3) A more elaborate description between the backbone technique GraNet and AST is required to help readers understand the proposed method more clearly.

(3) There are quite many typos and inconsistencies in the paper. A significant polishment is required to be published. i> ''s+r'' in Eq.2 should be  capital. ii> RigL and Rigl should be consistent. iii>  line 129, sub-net 1 < sub-net 2 < sub-net 3: the meaning of this comparison is vague. Does this mean sparsity or model size? iv> what is the difference between inner group and inter group? It would be better to add the inter group in Figure3 as well. v> figure2: I assume the sparsity of the subnet with different colors is different from each other, but green and blue have exactly the same number of connections. vi> the readability of the Tables in the paper is low. I suggest arranging different baselines according to their accuracy, e.g., in ascending order.

---

> ### Author Response · Authors · 2022-08-02
> **Response to Reviewer x1xA**
>
> **Q1**: Several naive baselines are missing.
>
> **A1**:  We thank the reviewer for pointing out the additional baselines. To provide the detailed analysis to the reviewer, we analyze the impact of fine-tuning based on the following three perspectives:
>
> - A short time of fine-tuning from the dense pre-trained model (as pointed out in Q1-(1)).
> - Start with sparse pre-training, fine-tune the high sparsity models with a short epochs, while keep the overall training cost~(time) as same as AST (as pointed out in Q1-(2)).
> - To further clarify the advantages of AST, we also investigate another perspective: a short time of fine-tuning from the sparse pre-trained model.
>
> Same as the experimental setup in the main paper, we conduct the experiments based on the wide ResNet-32 model on CIFAR-100 dataset. Given the dense pre-trained model, we separately fine-tune the dense model to achieve 90\%, 95\%, and 98\% sparsity with minimum efforts. As shown in Table 10 of the revision, fine-tuning from a dense model in a short period cannot achieve comparable accuracy as the proposed AST algorithm. Furthermore, the 160 epochs of pre-training and additional fine-tuning elevate the overall training costs.
>
> In addition to the dense model fine-tuning, we address the second concern of the reviewer by performing the sparse progressive training while keeping the overall training cost to be the same as a single AST training. With the wide ResNet-32 model, we first sparsify the model to 90\% sparsity from scratch with 60 epochs. Subsequently, we prune the 90\% sparse model to 95\% and then to 98\% sparsity with 50 epochs of fine-tuning. Compared to the single AST training, the total training effort is the same (60+50+50=160 epochs). As shown in Table 11 of the revision, such an individual pruning method failed to achieve the performance as the proposed AST training method. The large accuracy gap suggests the necessity of the proposed alternative sparsification training.
>
> On top of the Q1-(2), we further investigate the impact of fine-tuning based on a pre-trained sparse model. We first fully train a sparse subnet with 90\% sparsity (with 160 epochs) and prune the resultant model to 95\% and 98\% with a minimum amount of fine-tuning. As shown in Table 12 of the revision , fine-tuning the 90\% sparse model to 95\% or 98\% sparsity with up to 30 epochs cannot achieve comparable accuracy as AST, with even higher total training effort.
>
> The experimental results in Table 10, Table 11, and Table 12 of the revision suggest that it is difficult for individual fine-tuning to achieve the level of high sparsity and high accuracy as the proposed AST, regardless of the initial sparsity of the inherited model checkpoint.
>
> **Q2**: The accuracy of AST is relatively lower compared with GraNet on ImageNet.
>
> **A2**: First, we want to emphasize that we already achieve the best SOTA accuracy and training cost performance for training multiple sub-nets based dynamic network all-at-once. We provide comparison to GraNet to just show that our multi-sub-net training scheme could achieve the accuracy that is close to one sparse network. We agree with the reviewer that the insufficient update of each sub-net is one of the important reasons that cause accuracy degradation, since each sub-net cannot be fully trained by the entire training set inside each epoch. We have studied the effectiveness of the larger training time for the proposed method. As shown in Fig.8 of the original manuscript, compared to the individually trained GraNet baseline (total N=3$\times$, i.e., training 3 times of separated networks), the proposed method achieves the same accuracy in all three sparse models with only $\sim$2$\times$ averaged total training effort.
>
> **Q3**: A more elaborate description between GraNet and AST
>
> **A3**: GraNet is an in-training sparsity method that primarily focuses on training a single sparse model under a pre-defined sparsity ratio, generating a static and fixed sparse network. In contrast, AST trains multiple sparse sub-nets at one training procedure without extra training cost, compared to training a single sparse model independently. To generate and sparsely train each sub-net, we adapt the prune-and-regrow scheme from GraNet[1]. Specifically, AST starts from a randomly initialized sparse model. Then during the proposed alternatively training procedure, AST applies the prune-and-regrow mechanism in each training iteration with the sub-net-specific sparsity ratio. Besides, we also propose the gradient correction technique during alternative sparse training process to remove the conflicting gradient direction between sub-nets. We have added the corresponding modification in the revised manuscript.
> To further present the proposed AST method, we also added the pseudocode of the algorithm in the Appendix.
>
> **Q4**: There are quite many typos and inconsistencies.
>
> **A4**:  We have addressed all the typos and suggestions in the revised version.

---

> > ### Comment · Reviewer_x1xA · 2022-08-08
> > **Response**
> >
> > Thanks for the further explanation. My concerns have been addressed. I will increase my score to 6.

---

> > > ### Author Response · Authors · 2022-08-09
> > > **Many thanks for your further updates!**
> > >
> > > We thank the reviewer very much for further reviewing our response and increasing the score!

---

### Official Review · Reviewer_LUjM · 2022-07-10

**Rating:** 4
**Confidence:** 3
**Soundness:** 2 fair
**Presentation:** 3 good
**Contribution:** 2 fair

**Summary:**

This paper presents a sparse neural network training algorithm. The technical highlights include:

(1) The proposed algorithm adopts a scheme similar to Reptile, which sequentially trains several sparse sub-nets.

(2) To handle the conflicting direction of weight updates among sub-nets. Gradient correction is conducted to alleviate the negative gradient inner product.

(3)As for signal sparse subset training, two observations are reported that enabling the freedom of exploring unique architectures of each sub-net elevates the learning capacity of AST over the S-Net, and intermittent sparsity increment among sub-nets stabilizes AST process.

**Questions:**

1. If we view the training of each sub-network as a learning task. Is the overall framework a special case of Reptile?

2. The effect of gradient correction is not clear. What will the “negative gradient” and “conflicting direction of weight updates among sub-nets ” bring about to the learning process? I haven’t seen sufficient reason that it is necessary to deal with the issue. The experiment comparison is only conducted on two Cifar datasets.

3. The two observations in Sect. 3.3 are not sufficiently justified in the experiment.

4. Some critical technical details are missing. For example, is the final sparse network an ensemble of all trained sub-nets or the last sparse sub-net?

**Strengths And Weaknesses:**

Strengths:
1. The idea that learning sparse sub-nets sequentially and recursively is interesting.
2. The sparse network learned by the proposed algorithm has better performance than previous algorithms.
3. The paper writing is good, with sufficient related work introduction.

Weaknesses:
My major concerns are:
1. Experiments are kind of weak.
2. Some of the proposed technical points lack sufficient experiment verification.
3. The algorithm novelty.

---

> ### Author Response · Authors · 2022-08-02
> **Response to Reviewer LUjM**
>
> **Q1**: If we view the training of each sub-network as a learning task. Is the overall framework a special case of Reptile?
>
> **A1**: We thank you for raising this question. The proposed framework could be considered as a special case of Reptile with the following particularity: 1) as you mentioned, we treat each sub-network instead of data, as a task; 2) different from Reptile, which aims to learn a meta-model for fast adaption, the learning objective of this work is to learn an optimized model that includes multiple sparse sub-networks for dynamic inference.
>
> **Q2**: The effect of gradient correction is not clear. What will the “negative gradient” and “conflicting direction of weight updates among sub-nets ” bring about to the learning process? I haven’t seen sufficient reason that it is necessary to deal with the issue. The experiment comparison is only conducted on two CIFAR datasets.
>
> **A2**: First, we would like to mention that the `negative gradient' means the negative cosine similarity of the gradient between sub-nets. During the learning process of the proposed method, two sub-nets that have highly shared partial parameters will be updated at two consecutive iterations alternatively. In this case, if the gradient of the second sub-net has a negative cosine similarity with the first sub-net, the weight update of the second sub-net will have a conflicting direction with the first sub-net, leading to accuracy degradation. To avoid this issue, the proposed gradient correction technique removes the conflicting gradients within the inner-group iterations so as to reduce the weight updating interference between sub-nets. We believe that this technique is new and, for the first time, applied to training a dynamic network with a pool of different sub-nets. For the experiment comparison, we have demonstrated the effectiveness of the proposed gradient correction on both CIFAR-10 and CIFAR-100 datasets as shown in Table 2 of the original manuscript. Due to the limited rebuttal time, we couldn't obtain the comparison results on the ImageNet dataset yet. But we will include them in the next revised version.
>
> **Q3**: The two observations in Sect. 3.3 are not sufficiently justified in the experiment.
>
> **A3**:  We have the following response for the two observations:
> - Regarding the observation 1, we have summarized the comparison results between the investigated Completely-subset (CS) scheme and the Non-disjoint scheme (ND) in Table 1 of the original manuscript. As a result, on CIFAR-10 dataset, the ND scheme has the outperformed performance compared to the constrained completely-subset scheme (CS).
>
> - For the observation 2, to validate the effectiveness of the proposed extended adjustment method~(EA), we add an ablation study on various adjustment periods $\Delta \tau$, which is used to determine the frequency of sub-nets switching. As shown in the Table R1 below, compared to the smaller adjustment period (i.e., $\Delta \tau = 0, 90$), $\Delta \tau = 300$ achieves the best accuracy on all three sparsity levels. The reason is that performing sub-net switching frequently elevates the instability of model optimization.
>
> Table R1: The impact of the extended adjustment period. Given the wide ResNet-32 and CIFAR-100 dataset, sweep the sparsity update interval from 0 epoch up to 300 steps.
> |    Dataset    |                |     CIFAR-100 Acc (%)     |                |
> | :-----------: | :------------: | :-----------------------: | :------------: |
> |   ResNet-32   |                | Dense Model Acc. = 74.94% |                |
> | $\Delta \tau$ |      90%       |            95%            |      98%       |
> |       0       | 72.92$\pm$0.27 |      72.25$\pm$0.20       | 68.20$\pm$0.07 |
> |      90       | 73.28$\pm$0.13 |      72.72$\pm$0.20       | 68.25$\pm$0.03 |
> |      300      | 73.41$\pm$0.04 |      72.57$\pm$0.15       | 68.42$\pm$0.15 |
>
> **Q4**: Some critical technical details are missing. For example, is the final sparse network an ensemble of all trained sub-nets or the last sparse sub-net?
>
> **A4**: We thank you for raising this question. We would like to highlight that the main objective of this work is to learn a sparse super-network that includes multiple sub-nets for dynamic inference. In this case, the final learned sparse network is an ensemble of all trained sub-nets where the weights are partially shared with each other to enable run-time inference switching (i.e., dynamic inference).

---

> ### Author Response · Authors · 2022-08-08
> **Look forward to your feedback**
>
> Dear reviewer LUjM,
>
> Thanks again for your time. As the deadline for discussion is approaching, we really hope to have a further discussion with you to see if our response solves the concerns. We are happy to provide any additional clarifications that you may need.
>
> Best wishes, Author

---

> ### Author Response · Authors · 2022-08-09
> **Hope We Resolve Your Questions and Concerns**
>
> Dear reviewer LUjM,
>
> Thanks again for all your fruitful comments. We hope our detailed responses can resolve your questions and concerns, please kindly reconsider the overall score of our work
>
> Best wishes, Author

---

### Meta-Review · Area_Chair_eUnD · 2022-08-25

**Recommendation:** Accept
**Confidence:** Less certain

**Metareview:**

The authors presented an alternating sparse training method for building dynamic inference networks. The method comes with an extensive empirical study over multiple benchmark datasets showcasing its advantages over prior dynamic inference methods. The paper is generally well organized and clearly presented. The technical novelty, however, seems to be somewhat limited given that the inspiration of method and theoretical interpretations are drawn largely from the findings in Repitle for gradient based meta-learning. The reviewers also raised several other concerns regarding the connection to meta-learning, additional baselines and experiment details. The authors managed to address some of these in their responses.

After discussions, the reviewers reached a majority consensus that this is an interesting and technically sound paper where strengths outweigh weaknesses. Therefore, I recommend that paper can be accepted if room available and the promised revision is made to address the issues raised.


**Award:**

No

---

### Decision · Program_Chairs · 2022-09-14

Accept